# Simple Policy Gradients for Reasoning with Diffusion Language Models

**Anthony Zhan** [1]

## Abstract

Diffusion large language models (dLLMs) represent a promising alternative to autoregressive LLMs; however, the lack of effective post-training techniques, including reinforcement learning (RL), remains a key challenge for dLLMs, especially for downstream applications. Existing approaches often rely on a sequence-level view that requires biased likelihood approximations. In this work, we propose Amortized Group Relative Policy Optimization (AGRPO), a policy gradient algorithm that leverages the Markovian nature of dLLM generation, optimizing individual denoising steps rather than full sequences. Our approach improves the theoretical alignment between training and inference policies and also admits efficient, unbiased gradient updates via a novel timestep estimation scheme. We demonstrate AGRPO's effectiveness on different math and reasoning tasks, achieving absolute accuracy gains of +59.4% and +69.7% on Countdown and Sudoku over the base LLaDA model, exceeding comparable methods such as diffu-GRPO.

## 1. Introduction

Many recent efforts in LLM research have centered around reinforcement learning, specifically in the verifiable reward (RLVR) setting. In a typical setup, base models are trained on math or coding problems and incentivized to reason through the solution step-by-step, getting a reward if the final answer is correct. The main goal of RLVR is to elicit mathematical thinking/reasoning capabilities, allowing models to solve complex real-world tasks.

This wave of interest in RL and reasoning, initially spurred by models like OpenAI's o1 (OpenAI et al., 2024) and DeepSeek's R1 (DeepSeek-AI et al., 2025a), has led to numerous post-training algorithms designed specifically for autoregressive (AR) LLMs. With the success of these algorithms, chief among them Group Relative Policy Optimization (GRPO) (Shao et al., 2024), AR LLMs have grown incredibly strong on problem-solving benchmarks, with closed models even achieving gold medal performance at competitions such as the IMO and IOI, a remarkable feat (Luong & Lockhart, 2025; Lin & Cheng, 2025).

In a parallel line of research, diffusion language models have recently emerged as an alternative to the traditional autoregressive paradigm. Starting from the denoising/score matching framework that underlies continuous diffusion, works such as D3PM (Austin et al., 2021) and SEDD (Lou et al., 2024) show how to extend this framework to discrete settings like language. Successive efforts (Sahoo et al., 2024; Ou et al., 2025) have further simplified the theory, with most recent works settling on the "absorbing" or "masked" diffusion framework. We henceforth refer to this class of masked diffusion models as dLLMs.

Current open-source dLLMs, such as LLaDA (Nie et al., 2026) and Dream (Ye et al., 2025), are often on par with AR models such as LLaMA3-8B and Qwen2.5-7B on standard LLM benchmarks. Once trained, these models can go beyond standard left-to-right generation by conditioning on arbitrary masked sequences and sampling multiple tokens in parallel.

However, these models still struggle to match AR models in downstream tasks that require long-form thinking and reasoning. This discrepancy stems from fundamental challenges in designing post-training objectives for dLLMs: AR models have easy access to sequence-level likelihoods through AR factorization, whereas diffusion models must resort to ELBO-like approximation, resulting in biased gradient updates and overall suboptimal policy optimization.

Our work helps dLLMs close this gap by proposing a principled policy gradient algorithm designed for dLLMs: Amortized Group Relative Policy Optimization (AGRPO). Unlike existing sequence-level approaches, we first establish a finer step-level formulation of the post-training problem, tying it to the iterative denoising process used by dLLMs. Then, through a simple modification of the policy gradient objective — by viewing the inner sum as an expectation over timesteps — we show how to make training scalable for long-form reasoning tasks with hundreds of denoising steps.

[1]Stanford University. Correspondence to: Anthony Zhan <azhan9@stanford.edu>.

*Proceedings of the 43rd International Conference on Machine Learning*, Seoul, South Korea. PMLR 306, 2026. Copyright 2026 by the author(s).

Our main contributions are as follows:

- **Soundness.** We derive an unbiased policy gradient objective for dLLMs, rigorously grounded in a multi-step denoising MDP, explaining how our approach sidesteps the need for ELBO-like approximations.

- **Efficiency.** Through a new timestep estimation technique, we show how to implement our algorithm in a compute-efficient way that avoids costly full-trajectory updates, and discuss various practical tradeoffs.

- **Efficacy.** We train models on four reasoning tasks (GSM8K, MATH, Countdown, and Sudoku), showing that AGRPO consistently outperforms previous ELBO-based methods. Furthermore, models trained with AGRPO retain high accuracy even when sampled with much fewer steps, a unique capability of post-trained dLLMs.

## 2. Preliminaries

### 2.1. dLLM Pretraining

The most common form of discrete diffusion for language is the masked (or "absorbing") approach, where models are trained to reverse data corrupted by randomly masking tokens (Lou et al., 2024; Sahoo et al., 2024; Arriola et al., 2025). Concretely, given a distribution $p$ on sequences of discrete tokens $x = (x_1, \ldots, x_n)$, models are trained to maximize the following evidence lower bound (ELBO) on the likelihood (Nie et al., 2026; Ou et al., 2025):

$$\mathcal{L}(\theta) = \mathbb{E}_{\substack{t \sim U[0,1] \\ x, x^t \sim p^t}} \left[ \frac{1}{t} \sum_{x_i^t = \blacksquare} \log p_\theta(x_i \mid x^t) \right] \quad (1)$$

where $x, x^t \sim p^t$ means that $x$ is sampled from $p$ and $x^t$ is obtained from $x$ by independently setting each token $x_i$ to the mask token $\blacksquare$ with probability $t$. Similar to BERT (Devlin et al., 2019), the goal is for the model to learn marginal distributions of masked tokens conditioned on arbitrary context.

This masked token prediction objective can be viewed as a more general version of the next token prediction objective used to train AR models. As a result, dLLMs are more flexible than traditional LLMs: they can generate text in arbitrary order besides left-to-right, and can decode many tokens in parallel. A critical tradeoff of leaving behind AR factorization, however, is that the *sequence likelihood becomes intractable*: dLLMs cannot directly compute the likelihood of producing a given sequence. In particular, for dLLMs such as LLaDA, doing so would require marginalizing over $O(2^n)$ partially masked states, which is simply infeasible for most sequence lengths $n$.

### 2.2. dLLM Inference

To generate text, dLLMs start with an all- or partially-masked sequence, obtain marginal distributions for each masked token, and then unmask some of these by sampling from the marginal. The positions to be unmasked can be chosen either randomly (adhering to the reverse diffusion process) or by keeping the tokens with highest probability (as proposed by Nie et al. (2026)). We refer to these as "random" and "confidence-based" unmasking, respectively. The rest of the tokens are kept the same — masked tokens remain masked, previously unmasked tokens remain unmasked — and this new sequence is fed back into the model. This process is repeated until all tokens are unmasked.

Throughout this paper, we use $m$ to refer to the number of sampling steps, and $n$ to refer to the sequence length. A nice advantage of diffusion models is the ability to dynamically adjust the number of tokens unmasked per step $n/m$. Typically, the ratio $n/m$ is chosen to be relatively small ($\leq 4$) — unmasking more tokens at each step severely degrades output quality (Lou et al., 2024; Nie et al., 2026). We show in later sections that post-training actually changes this, allowing for much higher values of $n/m$ without clear degradation.

For more details on dLLM inference, see Appendix F.

### 2.3. Reinforcement Learning and MDPs

Markov decision processes (MDPs) are a formalization of sequential decision-making problems consisting of a state space $\mathcal{S}$, an action space $\mathcal{A}$, a transition kernel $p(s' \mid s, a)$, and a reward function $r(s, a)$. Broadly speaking, the goal of reinforcement learning (RL) is to learn a policy $\pi(a \mid s)$ that maximizes the expected return:

$$\arg\max_\pi \mathbb{E}_{\tau \sim \pi} \left[ \sum_{t=0}^{T} r(s_t, a_t) \right] \quad (2)$$

where $\tau$ represents a trajectory (or "rollout"), i.e., a sequence of states and actions $(s_0, a_0, s_1, a_2, \ldots, s_T, a_T)$ where $a_i \sim \pi(\cdot \mid s_i)$ and $s_{i+1} \sim p(\cdot \mid s_i, a_i)$.

### 2.4. Policy Gradient Methods

Policy gradients (PG) comprise a popular class of algorithms used to train neural network-parameterized policies $\pi_\theta$ to maximize expected return (2) (Sutton et al., 1999). The simplest form of policy gradients, REINFORCE (Williams, 1992), involves the following gradient update:

$$\nabla_\theta \mathcal{J}_{PG}(\theta) = \mathbb{E}_{\tau \sim \pi}$$
$$\left[ \left( \sum_{t=0}^{T} \nabla_\theta \log \pi_\theta(a_t \mid s_t) \right) \left( \sum_{t=0}^{T} r(s_t, a_t) \right) \right]. \quad (3)$$

More sophisticated algorithms such as Proximal Policy Optimization (Schulman et al., 2017) improve upon this objective by introducing a learnable reward baseline (i.e., a value function) and clipped importance sampling to allow for off-policy updates.

Further details aside, we highlight that the core structure of all PG objectives is the same: compute the likelihood of each action conditioned on state, perform a reward-weighted sum over the full trajectory, and backpropagate.

## 3. Policy Gradients for Diffusion Models

In this section, we describe the standard sequence-level framing of LLM post-training, and why such objectives are ill-suited for diffusion models. We then describe how the dLLM denoising process gives rise to a multi-step MDP, resulting in a stepwise PG objective with tractable likelihoods.

### 3.1. LLM Post-Training with Policy Gradients

In the RLVR setting, the typical way to describe the post-training problem as an MDP is as follows: states $s$ correspond to prompts $q$, actions $a$ correspond to model completions $x$, and the reward $r$ is provided by comparing $x$ to a ground truth answer. (Transitions are deterministic, so we can safely ignore the transition kernel $p$.)

Even though tokens are sampled sequentially, thanks to chain rule factorization, we can effectively view AR LLMs as policies over entire sequences. Under this view, the trajectory collapses to a single step $\pi_\theta(x \mid q)$, and the PG update (3) takes the convenient form

$$\mathbb{E}_{x \sim \pi(\cdot \mid q)}[\nabla_\theta \log \pi_\theta(x \mid q) r]. \tag{4}$$

This one-step PG formulation is quite effective on many LLM post-training tasks, including chain-of-thought reasoning; we discuss the large body of work devoted to this setting in Section 6.

As discussed in Section 2.1, however, diffusion models don't admit a tractable sequence-level likelihood. Simply replacing the likelihood $\log \pi_\theta(x \mid q)$ with an ELBO (as in pretraining) introduces several problems: biased and high-variance gradient updates, inability to correct negative-advantage traces, and a theoretical mismatch between the trained policy and the actual rollout-generating policy (Wang et al., 2026a; He et al., 2025).

Existing approaches such as diffu-GRPO and SPG (Zhao et al., 2025; Wang et al., 2026a) seek to address these issues via heuristically chosen masking processes or carefully tuned surrogate objectives. We therefore ask: *Is there a simpler, more principled approach without such compromises?*

### 3.2. Text Generation as a Multi-Step MDP

The previous sections establish that diffusion models are incompatible with the usual one-step MDP view of post-training. This motivates us to consider a different *multi-step* MDP formulation of the generation process, where

- the state $s_t$ is the partially masked sequence $x_t$,

- the actions $a_t$ are the $n/m$ tokens to be unmasked[1] at timestep $t$, and

- the verifiable reward $r$ is provided after the sequence is fully unmasked, i.e., at the last timestep.

In other words, a single step in this MDP is taken to be an individual unmasking step rather than the entire generation process.

This formulation, which aligns more naturally with how diffusion models are parameterized, has been adopted by previous works on RL for continuous diffusion such as DDPO and Flow-GRPO (Black et al., 2024; Liu et al., 2025a). Crucially, this perspective of denoising as a multi-step MDP allows for *true stepwise likelihoods* in the PG objective (3), sidestepping the need for lower bounds or approximations.

Treating dLLM generation as a multi-step MDP also naturally aligns the partially masked states between training and inference policies, resolving a key drawback of ELBO-based approaches. (This provides a principled justification for tricks such as block-wise masking that often improve performance in ELBO-based methods (Wang et al., 2026a).)

*Remark* 3.1. Recall that dLLM inference works by factorizing the joint probability of an unmasking step $\pi(x_t \mid x_{t-1})$ as the product of marginal token probabilities. While this stepwise likelihood is not an "exact" likelihood in the any-order AR sense (Ou et al., 2025), the PG objective does not require this "exact" definition anywhere, merely that actions are consistently defined between the training and inference policies, which holds as long as $n$ and $m$ are constant.

### 3.3. Larger Models and Longer Trajectories

Note that although the multi-step denoising perspective allows for tractable likelihoods, computing (3) naively requires a separate forward pass for every denoising step. In domains such as RL for robotics where the policy network $\pi_\theta$ is lightweight, this can be trivially parallelized on a GPU by batching many state-action pairs into a forward pass.

However, for large diffusion transformers which must attend to complex sequences, forward passes are often GPU

---

[1]For simplicity, we consider the unmasking positions to be exogenous, not something the model explicitly optimizes, although our framework cleanly extends to this case as well.

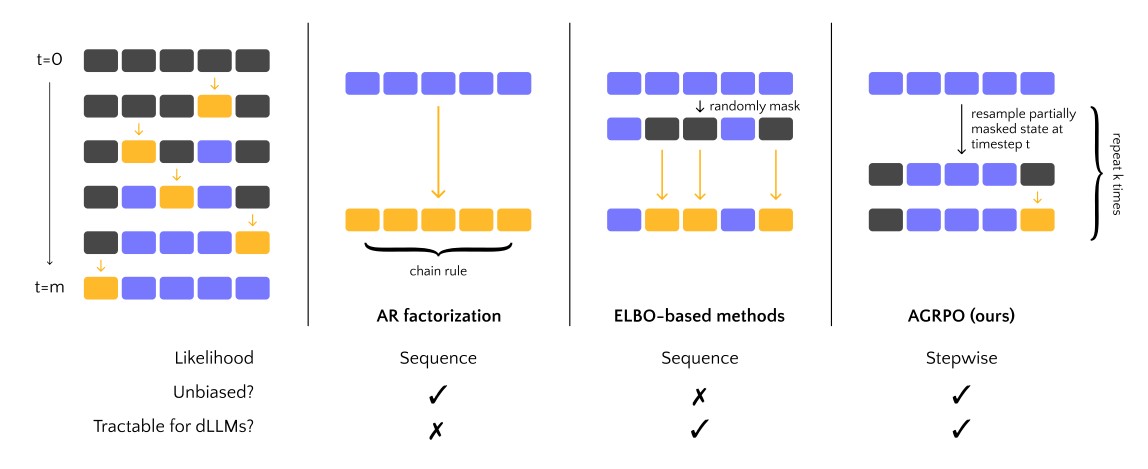

*Figure 1.* Existing RL post-training algorithms focus on sequence-level likelihoods and require either autoregressive factorization or ELBO-like bounds, which result in biased policy updates. Our proposed algorithm instead focuses on individual unmasking steps, aligning more naturally with the dLLM generation process.

compute-bound (i.e., batching increases the wall time linearly). For image generation tasks where the number of diffusion steps $m$ is small, spending $m$ forward passes to compute the full objective can work, but this becomes impractical for reasoning tasks with hundreds of steps. We propose a novel way of addressing this problem via *timestep estimation* in Section 4.1.

# 4. Amortized Group Relative Policy Optimization

Before presenting the AGRPO objective, we first introduce some notation. Let $D$ be a distribution over prompts $q$, $\{x^i\}_{i=1}^G$ a group of $G$ rollouts (or trajectories) conditioned on some $q$, and $r_i$ the respective rewards. Each $x^i$ has completion length $n$, and is generated with $m$ unmasking steps. We use $x_t$ to denote the partially masked state at timestep $t$. For example, $\pi_\theta(x_1 \mid x_0)$ represents the probabilities of the first $n/m$ tokens to be unmasked. (Recall that the unmasking order need not be left to right.)

## 4.1. From PPO to AGRPO

In this section, we show how to derive the AGRPO objective by reinterpreting the inner sum in the PG objective as an expectation across timesteps. Instead of the form given in Equation (3), we work with PPO (Schulman et al., 2017), a more modern form which includes advantages and importance sampling (and clipping, which we temporarily omit for clarity). With the notation above, the PPO surrogate objective is:

$$\mathcal{J}(\theta) = \mathbb{E}_{\substack{q \sim D \\ \{x^i\}_{i=1}^G \sim \pi_{\text{old}}(\cdot|q)}} \left[ \frac{1}{G} \sum_{i=1}^G \frac{1}{m} \sum_{t=1}^m \frac{\pi_\theta(x_t^i \mid x_{t-1}^i)}{\pi_{\text{old}}(x_t^i \mid x_{t-1}^i)} A_i \right] \quad (5)$$

where $A_i = r_i - \text{mean}\{r_i\}$ is the group-normalized advantage estimate and $\pi_{\text{old}}$ is the policy under which rollouts are sampled, which is updated every $\mu$ gradient steps.

Following recent works (Liu et al., 2025b), we avoid dividing by $\text{std}\{r_i\}$ to avoid bias from particularly easy or hard problems where advantages have low variance.

*Remark* 4.1. Although the inner sum in (5) resembles the inner sum in the GRPO objective (Equation 3 in Shao et al. (2024)), we note that these are derived from fundamentally different assumptions: (5) assumes a sequential $m$-step unmasking process, whereas GRPO starts from the one-step objective (4) and decomposes the sequence likelihood into a sum of token-level likelihoods via the chain rule.

Now let $T$ be a categorical random variable drawn uniformly from $\{1, \ldots, m\}$. Then an unbiased estimator of the inner term in Equation (5) is

$$\frac{1}{G} \sum_{i=1}^G \mathbb{E}_{T \sim \{1,\ldots,m\}} \left[ \frac{\pi_\theta(x_T^i \mid x_{T-1}^i)}{\pi_{\text{old}}(x_T^i \mid x_{T-1}^i)} A_i \right].$$

This enables *Monte Carlo (MC) sampling* of the original objective: simply draw $k \ll m$ timesteps, computing the stepwise likelihoods, and average the resulting terms. Since the likelihoods are now tractable, the resulting gradient estimate is *unbiased*, unlike existing works which substitute ELBOs for sequence likelihoods.

The full AGRPO objective with clipping and KL penalty is

$$\mathcal{J}(\theta) = \mathbb{E}_{\substack{q \sim D \\ \{x^i\}_{i=1}^{G} \sim \pi_{\text{old}}(\cdot|q)}} \left[ \frac{1}{G} \sum_{i=1}^{G} \mathbb{E}_{t \sim \{1,\ldots,m\}} \right.$$

$$\left. \left[ \min(\rho_t^i A_i, \text{clip}\left(\rho_t^i, 1-\varepsilon, 1+\varepsilon\right) A_i) - \beta D_{\text{KL}} \right] \right], \quad (6)$$

where

$$\rho_t^i = \frac{\pi_\theta(x_t^i \mid x_{t-1}^i)}{\pi_{\text{old}}(x_t^i \mid x_{t-1}^i)}$$

is the importance sampling ratio and $D_{\text{KL}}$ represents the KL divergence between policies at timestep $t$, i.e., $D_{\text{KL}}(\pi_\theta(\cdot \mid x_{t-1}^i) \| \pi_{\text{ref}}(\cdot \mid x_{t-1}^i))$. Note that our multi-step framing enables us to estimate $D_{\text{KL}}$ via MC sampling as well without relying on sequence-level approximations. To estimate $D_{\text{KL}}$, we use a corrected version of Schulman (2020)'s $k_3$ estimator

$$D_{\text{KL}}(p\|q) = \mathbb{E}_{x \sim p_{\text{old}}} \left[ \frac{p(x)}{p_{\text{old}}(x)} \left( \frac{q(x)}{p(x)} - \log \frac{q(x)}{p(x)} - 1 \right) \right]$$

which is used by DeepSeek-V3.2 (DeepSeek-AI et al., 2025b).

Algorithm 1 provides an overview of our proposed algorithm. For practical considerations, including how to efficiently retrieve partially masked states $x_t$ and compute backward passes, see Appendix E.

---

**Algorithm 1** Amortized Group Relative Policy Optimization (AGRPO)

**Require:** policy $\pi_\theta$, # sampling steps $m$, # MC samples $k$
  $\pi_{\text{ref}} \leftarrow \pi_\theta$
  **while** not converged **do**
    $\pi_{\text{old}} \leftarrow \pi_\theta$
    sample prompt $q \sim D$
    sample rollouts $\{x^i\}_{i=1}^{G} \sim \pi_{\text{old}}(\cdot \mid q)$
    compute advantages $\{A_i\}$
    **for** $\ell = 1$ **to** $\mu$ **do**
      $\widehat{J} \leftarrow 0$         {stores the MC estimate}
      **for** $j = 1$ **to** $k$ **do**
        sample $t \sim \{1,\ldots,m\}$ uniformly
        $\rho_t^i \leftarrow \frac{\pi_\theta(x_t^i|x_{t-1}^i)}{\pi_{\text{old}}(x_t^i|x_{t-1}^i)}$
        $\widehat{J} \leftarrow \widehat{J} + \sum_{i=1}^{G} \big[ \min \big( \text{clip}\left(\rho_t^i, 1-\varepsilon, 1+\varepsilon\right) A_i,$
                        $\rho_t^i A_i \big) - \beta D_{\text{KL}}(\pi_\theta \| \pi_{\text{ref}}) \big]$
      **end for**
      compute AGRPO estimate $\mathcal{J}(\theta) = \frac{\widehat{J}}{kG}$
      backpropagate and take gradient step w.r.t. $\theta$
    **end for**
  **end while**

---

## 4.2. Variance Reduction Techniques

For a fixed sampling budget $k$, a naive MC estimator would draw $k$ i.i.d. timesteps, compute the objective, and average the results. This gives unbiased gradient estimates, but for reasoning tasks where not all timesteps contribute equally to the final reasoning trace, this also introduces estimator variance (on top of the inherent variance in RL).

We would ideally like a minimal variance timestep estimator so that we can train stably with as few samples $k$ as possible. Here we propose two such ways of reducing variance: *entropy importance sampling* and *low-discrepancy sampling*. In Section 5.5, we empirically test their effects against the naive MC estimator. The full variance-reduced AGRPO estimator, with a proof of unbiasedness, can be found in Appendix A.

### 4.2.1. LOW-DISCREPANCY SAMPLING

To ensure that the model learns from a full range of intermediate states, we can replace i.i.d. samples by anti-correlated samples that collectively cover $\{1,\ldots,m\}$ while still being marginally distributed uniformly. This is known as low-discrepancy sampling, and is used in practice to lower training variance for both continuous and discrete diffusion models (Kingma et al., 2021; Sahoo et al., 2024; Zheng et al., 2025). We follow Zheng et al. (2025)'s discrete low-discrepancy sampler, which is detailed in Appendix D.

A desirable property of low-discrepancy sampling is that in the limit $k \to m$, the estimator recovers the original sum over the full trajectory. In practice, however, we observe that most of the performance gains can be captured by a fraction of the total number of steps $m$, with heavily diminishing gains as the sampling budget $k$ approaches $m$. We investigate this tradeoff induced by $k$ in more detail in Section 5.4.

### 4.2.2. ENTROPY IMPORTANCE SAMPLING

Inspired by Wang et al. (2025), who found that a minority of high-entropy tokens were responsible for the majority of performance gains in RLVR, we propose an entropy-based importance sampling scheme.

Instead of drawing $T \sim \{1,\ldots,m\}$ uniformly, we compute an entropy score $e_t$ for each timestep $t$ by summing the entropies of the unmasked tokens at $t$. Then we draw $T$ according to $\Pr[T = t] \propto e_t$, compute the PG estimate at timestep $T$, and multiply by a correction term $\sum e_t/(me_T)$. This prioritizes updates from pivotal tokens that influence the reasoning trace, reducing the risk of missing rare but critical steps.

*Table 1.* Accuracies for different RL post-training methods across different reasoning tasks and generation lengths. For all diffusion models, outputs of length $n$ are generated with $m = n/2$ steps. Few-shot examples are denoted in parentheses, and the best accuracy for each task is **bolded**. Models trained with LoRA (rather than full fine-tuning) are denoted with †.

| Model (AR) | GSM8K | MATH |
| --- | --- | --- |
| DeepSeekMath-Base 7B | 64.2 | 36.2 |
| CoT SFT | 82.9 | 46.8 |
| CoT SFT + GRPO | **88.2** | **51.7** |

| Model \ Seq len | GSM8K (0) | | MATH-500 (0) | | Countdown (0) | | Sudoku (3) | |
| --- | --- | --- | --- | --- | --- | --- | --- | --- |
| | 256 | 512 | 256 | 512 | 256 | 512 | 256 | 512 |
| LLaDA-8B-Instruct | 79.9 | 78.2 | 36.6 | 37.4 | 20.3 | 27.3 | 25.5 | 18.8 |
| LLaDA 1.5 (VRPO) | 81.4 | 81.1 | 35.4 | 40.6 | 27.0 | 27.3 | 23.5 | 20.1 |
| diffu-GRPO† | 79.8 | 81.9 | 37.2 | 39.2 | 31.3 | 37.3 | 32.5 | 29.3 |
| wd1† | 80.8 | 82.3 | 34.4 | 39.0 | 51.2 | 46.1 | 32.1 | 22.5 |
| SPG† | 86.1 | 84.5 | **40.0** | 41.8 | 70.7 | 70.3 | 94.0 | 93.1 |
| AGRPO (ours)† | **88.0** | **88.1** | 39.4 | **42.0** | 79.7 | 82.4 | 95.2 | 95.2 |

## 5. Experiments

To empirically validate our proposed algorithm, we start from the open source base model LLaDA-8B-Instruct (Nie et al., 2026) and fine-tune models using AGRPO on four different reasoning tasks: GSM8K, MATH, Countdown, and Sudoku.

### 5.1. Datasets

GSM8K/MATH are standard problem-solving benchmarks consisting of 8.5k/12.5k math problems at the grade school/high school level, respectively (Cobbe et al., 2021; Hendrycks et al., 2021). Countdown is a popular math reasoning task where the model is given a list of 3-4 numbers and a target number; the goal is to combine the numbers using arithmetic operations $(+, -, \times, /)$ and parentheses to reach the target number (Pan et al., 2025). Sudoku is a planning task where the goal is to fill in a 4x4 grid of numbers according to uniqueness constraints.

These four tasks form a common set of benchmarks for the growing literature on dLLM reasoning (Wang et al., 2026a; Zhao et al., 2025; Tang et al., 2025). For consistency, we use the Countdown and Sudoku splits provided in SPG's codebase (Wang et al., 2026a). We use HuggingFace's `math-verify` library for parsing GSM8K, MATH, and Countdown answers.

### 5.2. Experimental Setup

Following previous dLLM RL works, we use Low-Rank Adaptation (Hu et al., 2021) instead of full fine-tuning. We fix the response length at $n = 384$ and the number of steps at $m = 128$ to balance wall time and coherence, both of which are important for RLVR, and we use $k = 24$ MC sam-

ples. (Both the rollout-generating policy and training policy use the same value of $m$, so the stepwise likelihoods are consistent.) Notably, despite being trained on a single configuration, we observe that model performance generalizes to different output lengths and steps.

During training, we generate rollouts with temperature 0.6 and random remasking to inject stochasticity and incentivize exploration. Random remasking makes additional sense in the context of AGRPO since it allows the model to learn from a diversity of intermediate states, even if the final state is the same.

Models are trained until convergence is observed (e.g., via the reward curve plateauing); we select checkpoints from the last 150 steps for testing and report the best accuracy among those checkpoints. For evaluation, we switch to confidence-based unmasking (which can be seen as a form of annealing (Nie et al., 2026)) with temperature 0 in order to keep the generation process consistent with previous works. Other hyperparameters, including $G$ and $\varepsilon$, can be found in Appendix B.1.

*Remark* 5.1. Although our algorithm includes a KL penalty, we find empirically that setting $\beta > 0$ does not yield any measurable benefits (and incurs an additional forward pass per gradient step), consistent with other RLVR works.

### 5.3. Results

**Strong reasoning improvements.** We report accuracies on test splits in Table 1. For diffusion models, AGRPO achieves the highest accuracy across all but one task, outperforming the base LLaDA model and other dLLM post-training methods, including diffu-GRPO (Zhao et al., 2025), VRPO (Zhu et al., 2025), and SPG (Wang et al., 2026a). At sequence length $n = 512$, we improve upon the previ-

ous best reported results by $+1.4\%$ on GSM8K, $+0.2\%$ on MATH-500, $+12.1\%$ on Countdown, and $+2.1\%$ on Sudoku. Baseline comparisons are discussed in greater detail in Appendix B.3.

Our results suggest that tailoring towards the Markovian nature of the diffusion process is the right way to extend AR LLM reasoning abilities to dLLMs, dispensing with the need for ELBO-like bounds.

**Comparable gains to GRPO.** Although performance on MATH still lags behind autoregressive models, we are able to achieve parity with DeepSeekMath-RL on GSM8K, which represents a substantial improvement for dLLMs. Additionally, we observe increased performance deltas for AGRPO compared to GRPO (e.g. for GSM8K, GRPO results in $+5.3\%$, whereas AGRPO results in $+9.9\%$ at $n = 512$).

### 5.3.1. INFERENCE TRADEOFFS

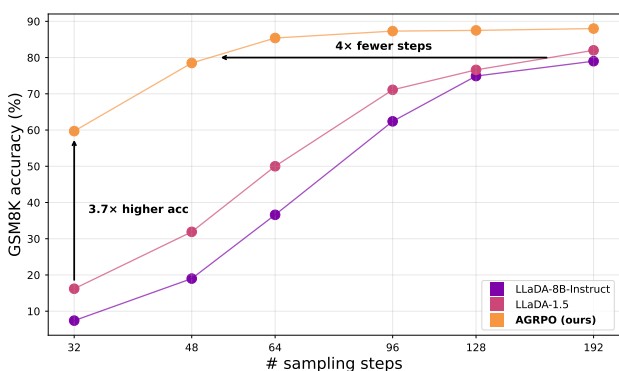

*Figure 2.* The compute/quality frontier for the GSM8K test split with response length $n = 384$. Lines show the possible tradeoffs at inference time for a specific model.

A key quality of dLLMs is their ability to trade off compute and quality at inference time. We examine how AGRPO affects the inference compute/quality frontier by evaluating models on the GSM8K test split at a fixed response length $n = 384$ and varying the number of sampling steps $m$. As shown in Figure 2, not only does the AGRPO model consistently achieve higher performance across all sampling steps, it matches baselines with *4x fewer sampling steps*, a substantial speedup. At the fastest setting, $m = 32$, AGRPO achieves 59.7% accuracy while sampling 12 tokens per step, an 8.1x performance improvement over LLaDA-8B-Instruct and 3.7x improvement over LLaDA 1.5. These results demonstrate that the reasoning skills instilled by AGRPO are *robust*, generalizing to different inference configurations despite being trained on a fixed $n$ and $m$. We give sample responses from this experiment in Appendix G.

To our knowledge, this is one of the first investigations of fine-tuned dLLMs where the checkpoint and context length

are *fixed* and the number of sampling steps is varied extensively. This opens up a new perspective on post-training dLLMs: by paying a fixed, upfront cost to train on a specific dataset, we can amortize the inference costs of future queries. This idea of few-step sampling as a learnable capability is similar to existing work on few-step distillation, and is a distinct benefit of dLLMs that standard AR models cannot realize.

### 5.4. Ablations on $k$

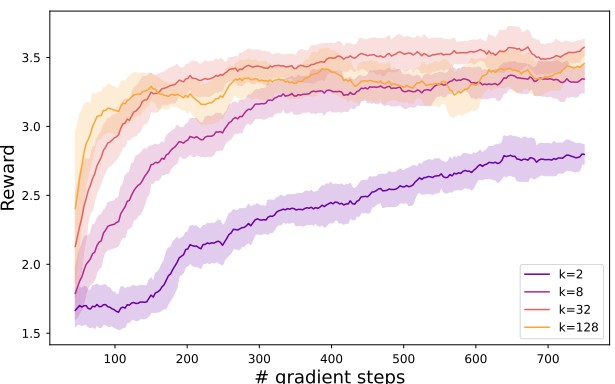

*(a)* GSM8K reward over training runs with different $k$. Shaded area represents intra-run variance over a rolling window of 15 steps.

| | **Average wall time (s)** | | | **Accuracy** |
|---|---|---|---|---|
| $k$ | Inference | Loss/backprop | Step | |
| 2 | 42.9 | 1.4 | 44.8 | 83.6 |
| 8 | 46.6 | 6.7 | 53.7 | 84.4 |
| 32 | 43.7 | 28.2 | 72.4 | 86.6 |
| 128 | 45.2 | 112.4 | 158.1 | 84.0 |

*(b)* Average wall time per training step and final accuracy for different components of AGRPO. Values are reported on 8xH100 GPUs with a global batch size of 128.

*Figure 3.* Reward curve and wall time comparisons for different values of $k$ on GSM8K with $n = 384$ and $m = 128$.

With Monte Carlo estimation, increasing the sampling budget by some factor $c$ reduces the variance by $c$. This is the simplest lens through which to understand $k$: increasing $k$ buys higher quality gradients at the price of more compute per step. In the context of online RL, however, the tradeoffs around $k$ becomes slightly more nuanced for two reasons: inference bottleneck and performance saturation.

**Rollouts are the bottleneck.** For reasoning tasks that require long chain-of-thought trajectories with hundreds of denoising steps, the training cost is mostly dictated by rollout generation, not the policy update itself. This effect is exacerbated in dLLMs due to their lack of KV caching. Consequently, a linear increase in $k$ corresponds to a linear increase in loss/backprop cost, but not total cost per step.

**Reward saturates early.** RL performance is controlled by the reward, so once the model has reached a certain reward level, performance stops improving and training has effectively converged. The number of steps to convergence depends on the learning signal per step, so it makes sense from a cost standpoint to spend more compute per step if it reduces the total steps.

As seen in Figure 3, our empirical findings are consistent with our expectations: larger values of $k$ increase the wall time per step but converge faster. Note that although $k = 32$ spends almost 2x more compute per step than $k = 2$, it takes far more than 2x as many steps for $k = 2$ to reach the same reward level. When we increase $k$ from 32 to 128, the compute per step more than doubles, but the reward only increases marginally faster and final performance actually degrades.

To summarize, choosing $k$ to be a healthy fraction of $m$ (here $k \approx m/4$) is optimal: if $k$ is too small, the learning signal isn't effective and inference costs dominate, and if $k$ is too large, the training cost quickly explodes while convergence is only slightly faster. Pinpointing the exact relationship between $n$, $m$, and the optimal value of $k$ is an area we identify for future work.

We give an informal argument for how these results can be anticipated in Appendix B.2 and similar wall time benchmarks against diffu-GRPO and SPG in Appendix C.

### 5.5. Ablations on Variance Reduction Techniques

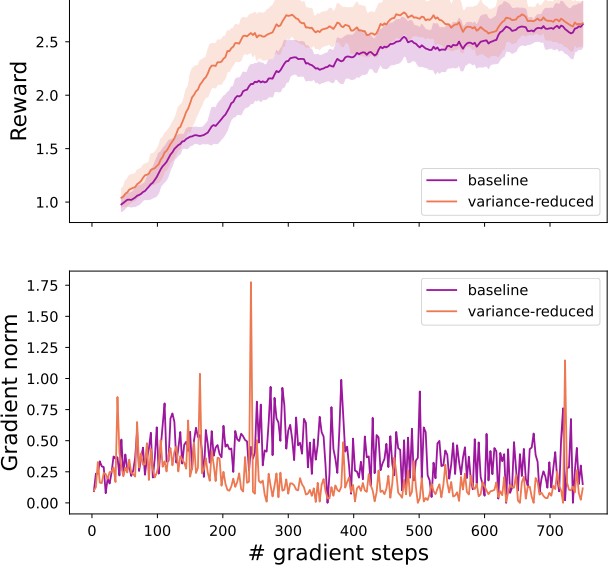

*Figure 4.* Reward and gradient norm plots for baseline and variance-reduced versions of AGRPO on Countdown with $n = 128$ and $m = 64$. The shaded area of the reward plot represents a rolling standard deviation with a window of 15 steps.

In this section, we investigate whether the variance reduction techniques proposed in Section 4.2 truly help empirically. We train two models, one with low-discrepancy sampling and entropy importance sampling and one without, on the Countdown task. All other hyperparameters, including the sequence length $n = 128$ and sampling steps $m = 64$, are the same between the two runs.

The reward curves and gradient norms are shown in Figure 4. Although both models reach a similar final reward level, the variance-reduced model converges faster, with fewer signs of early instability. In addition, the gradient norms for the variance-reduced model are consistently lower than the baseline throughout training (save a few outlier steps). These results suggest that low discrepancy sampling and entropy-based importance sampling do indeed provide less noisy, more valuable updates.

## 6. Related Works

**Policy gradient methods for LLMs.** Early attempts at RL with LLMs (Ouyang et al., 2022; Ziegler et al., 2020) focused on alignment with human preferences using manually-labeled preference data and methods such as PPO or Direct Preference Optimization (Rafailov et al., 2023). More recently, RL efforts have focused on reasoning capabilities, specifically for domains with verifiable rewards such as math and coding. LLMs post-trained with algorithms such as GRPO exhibited improved reasoning abilities beyond SFT (Luo et al., 2025) and even emergent capabilities such as backtracking and self-correction (Xiong et al., 2025). Subsequent works address issues related to response length bias, sample efficiency, and stability (Yu et al., 2025; DeepSeek-AI et al., 2025b).

**RL for continuous diffusion.** Continuous diffusion models are highly popular in areas like image generation (Sohl-Dickstein et al., 2015; Peebles & Xie, 2022), video generation (Ho et al., 2022), and robotic control (Chi et al., 2024). To better align these models with desired downstream behavior, e.g. text-to-image tasks, different PG methods for diffusion have been proposed, including DDPO (Black et al., 2024) and DPPO (Ren et al., 2024). These approaches formulate the RL problem as a multi-step denoising MDP, which we show can be successfully extended to dLLMs.

**RL for dLLM reasoning.** Discrete diffusion models allow the continuous diffusion framework to be extended to areas such as text and DNA sequences (Lou et al., 2024; Austin et al., 2021). Post-training these models remains an active area of research; while some works focus on the more general continuous time/score entropy perspective (Zekri & Boulle, 2025), many works choose to focus on masked diffusion models such as LLaDA, which are more similar to traditional LLMs. Approaches to dLLM RL can be broadly

divided into two groups: ELBO-based approaches that rely on approximations of the sequence likelihood, such as d1 (Zhao et al., 2025), wd1 (Tang et al., 2025) and SPG (Wang et al., 2026a), and trajectory-based approaches that rely on stepwise likelihoods, such as TraceRL (Wang et al., 2026b) and MDPO (He et al., 2025). Our work is more similar to the latter, but with key statistical techniques that enable *scalable* training in reasoning settings while maintaining unbiased policy updates and strong downstream performance. We discuss these differences in more detail in Appendix C.

## 7. Conclusion

This work presents AGRPO, a policy gradient algorithm designed for dLLMs, grounded in the multi-step denoising perspective of RL. Unlike previous works that rely on sequence-level likelihood bounds, AGRPO uses tractable stepwise likelihoods, allowing for unbiased policy updates and better train/inference alignment. A key contribution is showing how to compute these updates in a practical, scalable way via timestep estimation and novel variance reduction techniques. Using our proposed algorithm, we post-train dLLMs on multiple reasoning tasks, beating comparable RL methods and redefining the inference compute/quality frontier. Our results demonstrate that policy gradient RL techniques can be effectively extended to the dLLM setting without the need for biased approximations; we hope future works can build on our methods, either theoretically or empirically, and further close the gap between dLLM and AR LLM post-training.

## Acknowledgments

We would like to thank Aaron Lou, Jiaqi Han, and Michael Li for their insightful discussions, especially during the early stages of this project. We also gratefully acknowledge Modal for providing compute support.

## Impact Statement

This paper presents work whose goal is to advance the field of Machine Learning. There are many potential societal consequences of our work, none which we feel must be specifically highlighted here.

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

# A. Proof of Unbiased Estimator

Using the notation of Section 4.1, define the entropy score $e_t^i$ as the Shannon entropy of the policy distribution $\pi(\cdot \mid x_{t-1}^i)$. (For dLLMs, this is the sum of the marginal entropies of the unmasked positions at step $t$.) Let $\bar{e}^i = \{e_t^i / \sum e_t^i\}_{t=1}^m$ be the vector of normalized entropy scores, and suppose $T^i \sim \mathrm{Cat}(\bar{e}^i)$. Then the full AGRPO estimator (with clipping removed for clarity) is

$$\widehat{\mathcal{J}}(\theta; T) = \frac{1}{G} \sum_{i=1}^{G} \frac{1}{m\bar{e}_{T^i}^i} \frac{\pi_\theta(x_{T^i}^i \mid x_{T^i-1}^i)}{\pi_{\mathrm{old}}(x_{T^i}^i \mid x_{T^i-1}^i)} A_i.$$

**Theorem A.1.** $\mathbb{E}[\widehat{\mathcal{J}}(\theta; T)] = \mathcal{J}(\theta)$, where $\mathcal{J}(\theta)$ is defined in Equation (5) and the expectation is taken w.r.t. randomness of $T = \{T_i\}_{i=1}^G$.

*Proof.* By linearity of expectation,

$$\mathbb{E}[\widehat{\mathcal{J}}(\theta; T)] = \frac{1}{G} \sum_{i=1}^{G} \mathbb{E}_{T^i} \left[ \frac{1}{m\bar{e}_{T^i}^i} \frac{\pi_\theta(x_{T^i}^i \mid x_{T^i-1}^i)}{\pi_{\mathrm{old}}(x_{T^i}^i \mid x_{T^i-1}^i)} A_i \right].$$

Since each $T^i$ is a categorical random variable, we can write out the inner expectations explicitly:

$$\begin{aligned}
\mathbb{E}[\widehat{\mathcal{J}}(\theta; T)] &= \frac{1}{G} \sum_{i=1}^{G} \sum_{t=1}^{m} \Pr[T^i = t] \frac{1}{m\bar{e}_t^i} \frac{\pi_\theta(x_t^i \mid x_{t-1}^i)}{\pi_{\mathrm{old}}(x_t^i \mid x_{t-1}^i)} A_i \\
&= \frac{1}{G} \sum_{i=1}^{G} \frac{e_t}{\sum_{t'=1}^m e_{t'}} \frac{\sum_{t'=1}^m e_{t'}}{me_t} \sum_{t=1}^{m} \frac{\pi_\theta(x_t^i \mid x_{t-1}^i)}{\pi_{\mathrm{old}}(x_t^i \mid x_{t-1}^i)} A_i \\
&= \frac{1}{G} \sum_{t=1}^{m} \frac{1}{m} \sum_{i=1}^{G} \frac{\pi_\theta(x_t^i \mid x_{t-1}^i)}{\pi_{\mathrm{old}}(x_t^i \mid x_{t-1}^i)} A_i = \mathcal{J}(\theta).
\end{aligned}$$

$\square$

*Remark* A.2. There are many potential unbiased ways to estimate the PG objective with Monte Carlo sampling; entropy-weighted timestep estimation is a natural way, but there could be e.g. different importance sampling schemes to better reduce variance and accelerate training. We leave such explorations for future work.

# B. Experiment Details

### B.1. Training

Models are trained with LoRA rank $r = 64$ on 8xH100 GPUs for 400-900 steps (depending on the task). We use Hugging Face's TRL framework (von Werra et al., 2020) to implement the training code. Rewards are a combination of rule-based formatting rewards, i.e. the presence of `<think>` and `<answer>` tags, as well as binary ground-truth rewards (except for Sudoku, where the reward is the fraction of empty cells filled in correctly, which is dense).

For batched rollouts, we use a group size of $G = 8$ per prompt and 16 prompts per update, resulting in a global batch size of 128. The clipping threshold is set to $\varepsilon = 0.2$. The number of gradient updates per batch of rollouts $\mu$, which controls the "off-policyness" of the algorithm, is set to $\mu = 1$. As mentioned previously, we also assume $\beta = 0$, so the algorithm is fully on-policy with no KL penalty. Consistent with Yu et al. (2025) and Tang et al. (2025), we don't observe any drawbacks in terms of convergence speed or stability with this configuration.

### B.2. Training Cost Analysis

Continuing the discussion in Section 5.4, we show that under some simplifying assumptions, the total cost of RL training as a function of $k$ takes a quasiconvex form. For a given sampling budget $k \in \{1, \ldots, m\}$, suppose that the standard deviation of the MC estimator is $\sigma \propto 1/\sqrt{k}$, and as a crude approximation, that the learning signal per step is $\propto \sqrt{k}$, i.e., inversely proportional to $\sigma$. Define convergence as the model reaching some predefined learning threshold.

**Proposition B.1.** *Let $f(k)$ be the total cost of training a model with AGRPO until convergence. Then*

$$f(k) \approx \frac{C}{\sqrt{k}} + D\sqrt{k}$$

*for constants $C$ and $D$. Thus $f$ is quasiconvex in $k$, attaining a global minimum at some $1 < k < m$.*

*Proof.* Let the learning signal per step be $a\sqrt{k}$ and the convergence threshold be $b$. Then the number of steps needed is $b/(a\sqrt{k})$. The cost of each step is affine in $k$: $c + dk$, where $c$ is the cost of generating rollouts and $d$ is the cost of each MC estimate. The total cost is then the product of these terms:

$$\frac{b}{a\sqrt{k}}(c + dk) = \frac{bc}{a\sqrt{k}} + \frac{bd}{a}\sqrt{k}.$$

$\square$

### B.3. Evaluation

For all tasks and train/test splits, we use open source datasets from HuggingFace or GitHub. Accuracies for GSM8K are reported on the GSM8K-Platinum (Vendrow et al., 2025) test split, a cleaned version of the original GSM8K test split. Surprisingly, we found that models trained on GSM8K problems achieved higher accuracy on MATH-500 test problems than models trained on the MATH training split. The accuracy reported in Table 1 is thus from the GSM8K model; models trained on MATH showed more modest gains, achieving 38.0% at $n = 256$. We hypothesize that the more difficult nature of MATH, combined with the limited context window and weak priors of the base LLaDA model, made it harder to sample successful reasoning traces during RL.

Results for diffu-GRPO, wd1, and SPG are taken from the respective papers (Zhao et al., 2025; Tang et al., 2025; Wang et al., 2026a). We note that our reproduced baselines for LLaDA-8B-Instruct and LLaDA 1.5 are slightly higher than baselines reported in the aforementioned works (and that variations exists between these works as well). We attribute this to our use of a more amenable system prompt and robust parsing system (via symbolic libraries), which we applied consistently across all tasks to ensure a fair comparison.

For completeness, we provide a full comparison of reported LLaDA-8B-Instruct baselines against Zhao et al. (2025) and Wang et al. (2026a) in Table 2.

*Table 2.* Comparison of base model accuracies reported in previous work vs. reproduced in our work.

| Work | Model \ Seq len | GSM8K 256 | GSM8K 512 | MATH-500 256 | MATH-500 512 | Countdown 256 | Countdown 512 | Sudoku 256 | Sudoku 512 |
|---|---|---|---|---|---|---|---|---|---|
| d1 (Zhao et al., 2025) | LLaDA-8B-Instruct | 76.7 | 78.2 | 32.4 | 36.2 | 19.5 | 16.0 | 6.7 | 5.5 |
| SPG (Wang et al., 2026a) | LLaDA-8B-Instruct | 77.2 | 79.8 | 32.4 | 34.6 | 16.8 | 16.8 | 27.7 | 26.2 |
| AGRPO (ours) | LLaDA-8B-Instruct | 79.9 | 78.2 | 36.6 | 37.4 | 20.3 | 27.3 | 25.5 | 18.8 |

## C. Comparisons to Previous dLLM Post-Training Methods

Here we continue our discussion of dLLM post-training methods presented in Section 6.

**wd1.** Tang et al. (2025) build on diffu-GRPO by leveraging the closed-form solution $\pi^*$ of reverse KL-regularized policy optimization, namely $\pi^*(a \mid s) \propto \pi_{\text{ref}}(a \mid s)\exp(r(s, a)/\beta)$ where $\beta$ is the reverse KL coefficient. They use this to rewrite the PG objective in way that simplifies the importance sampling ratio $\pi_\theta(\cdot \mid q)/\pi_{\text{old}}(\cdot \mid q)$. However, their approach inherits from d1 the same reliance on ELBO-based likelihood approximations, limiting practical effectiveness.

**VRPO.** Introduced by Zhu et al. (2025) as part of the training process for LLaDA 1.5, Variance-Reduced Policy Optimization follows a similar approach to DPO (Rafailov et al., 2023), where the model directly learns from pairwise preference data without external rewards. The VRPO objective consists of the DPO objective with all sequence likelihoods replaced by

ELBOs, along with several tricks to reduce variance. Overall, the LLaDA 1.5 paper studies a slightly different RL setting (alignment/RLHF instead of reasoning/RLVR) and continues to rely heavily on likelihood bounds.

**DCoLT.** Inspired by autoregressive chain-of-thought methods, Huang et al. (2026) optimize a Diffusion Chain of Lateral Thought. Similar to our approach, they adopt a multi-step denoising view where actions consists of individual unmasking steps rather than the final sequence. However, there are several key differences that make AGRPO training much simpler: DCoLT requires training a separate module to rank which positions to unmask (according to a Plackett-Luce model) and accumulates gradients over *all $m$ timesteps* instead of using MC sampling to estimate gradients efficiently.

**SPG.** Most recently, Wang et al. (2026a) propose Sandwiched Policy Gradients, which combines an ELBO-like lower bound on likelihood with an evidence upper bound (EUBO). They identify an issue with PG objectives that doesn't appear in dLLM pretraining, namely the fact that likelihoods can be multiplied by a negative advantage estimate, which breaks the validity of optimizing a lower bound. By optimizing the EUBO in place of the standard ELBO for negative advantage trajectories, they achieve stronger empirical results (although their policy updates remain biased due to the sequence-level framing).

### C.1. Wall Time Comparisons

Since AGRPO involves multiple forward passes per gradient step, it incurs a higher wall-clock cost per update than ELBO-based methods like diffu-GRPO and SPG (see Table 3). However, because AGRPO updates the policy based on a wide range of intermediate states rather than only the final sequence, it provides a denser learning signal and therefore improved sample efficiency (i.e. requires fewer rollouts). Empirically, we observe rapid convergence to high reward levels across all tasks (see e.g. Figures 3a and 4), suggesting that the additional compute per step is offset by higher quality gradient updates.

*Remark* C.1. As previously emphasized, the main bottleneck in online RL with dLLMs is the inference time. Thus the additional cost of AGRPO's repeated sampling is less significant than it may appear; the increase in FLOPs per step is commensurate with modest increases in context length or batch size, which are commonly used to trade compute for optimization stability.

*Table 3.* Average wall time per gradient step for different dLLM RL methods on a controlled GSM8K training run with $n = 256$, $m = 128$, $G = 4$, and $\mu = 2$. Values are reported on 2xA6000 Ada GPUs with a global batch size of 16.

| | **Average wall time (s)** | | |
|---|---|---|---|
| **Method** | Inference | Loss/backprop | Step |
| diffu-GRPO | 42.4 | 0.5 | 43.8 |
| SPG | 41.9 | 1.0 | 44.6 |
| AGRPO ($k = 8$) | 41.6 | 10.4 | 53.3 |

## D. Low-discrepancy Sampling Details

An outline of Zheng et al. (2025)'s discrete low-discrepancy sampler is given below. The goal is draw $k$ samples from $T \sim \{0, \ldots, m - 1\}$.

1. Sample $U_0, \ldots, U_{k-1}$ i.i.d. from $\text{Unif}([0, 1])$.

2. "Bin" them into $k$ disjoint bins by defining $U'_j = (U_j + j)/k$ for $j = 0$ to $k - 1$.

3. Define the final set of $k$ samples $\{T_j\}_{j=0}^{k-1}$ as $T_j = \lfloor mU'_j \rfloor$.

Intuitively, the sampler divides the interval $[0, 1]$ into $k$ equal subintervals, samples from each one uniformly and shuffles the results, and then scales by a factor of $m$ and floors. In AGRPO, this allows us to "cover" a wide range of timesteps from 1 to $m$ when estimating the PG objective. We can also extend this to the batch level: instead of sampling $k$ timesteps, we sample $bk$ timesteps where $b$ is the batch size.

To use low-discrepancy sampling in conjunction with entropy importance sampling, we replace Step 3 by an inverse CDF transform to turn samples from $\text{Unif}([0, 1])$ into categorical samples.

# E. Practical Considerations

In this section, we discuss in detail various decisions and tradeoffs made regarding the actual implementation of AGRPO. As with all online RL algorithms, the goal of any implementation is to run as efficiently as possible while maximizing efficacy. We plan to release training and evaluation code publicly to showcase implementation details and encourage reproducibility.

**Caching partially masked states.** In order to obtain stepwise likelihoods, we must recreate the exact state at the step where that token was unmasked. To do this efficiently, we cache the unmasking order during generation so that each token is associated with a timestep $t$. Then, to get the partially masked state at timestep $t$, we simply mask out all tokens with timestep $\geq t$.

**Memory-efficient gradient accumulation.** Naively computing the AGRPO objective (6) and backpropagating would require keeping $k$ forward passes simultaneously in memory, which can quickly saturate GPU memory for 8B scale models. Instead, one can accumulate the gradient immediately after each MC sample is computed by calling `loss.backward()` (without taking an optimizer step) *inside* the for loop. This frees the computational graph and avoids excess memory usage.

**float64 Gumbel-based categorical sampling.** dLLMs typically use the Gumbel-max trick to sample from output logits. However, Zheng et al. (2025) point out that naively using `float32` causes an inconsistency between theoretical and actual behavior due to floating-point precision. We follow their recommendation of using `float64` for the sampling stage.

**Handling EOS tokens.** Since dLLMs generate with a fixed number of sampling steps, in later timesteps, the model can spend many "garbage" steps producing `EOS` tokens at the end of a sequence (while other sequences in the batch are still generating useful tokens). Gradient updates on these steps don't provide meaningful information to the model, so we set the max timestep in our low-discrepancy sampler (Section D) to be the last timestep where a non-`EOS` token was generated.

**Training/inference mismatches.** Yao et al. (2025) observe that for RL with traditional LLMs, mismatches between the inference engine and training code can cause subtle "off-policyness" and instability issues, especially when combined with bf16 precision. For AGRPO, we deploy a custom inference engine for LLaDA, making sure to track things such as the top-p/top-k sampling masks and reuse them in training, as suggested by DeepSeek-AI et al. (2025b).

# F. Remarks on dLLM Inference

Despite a more complicated training setup, dLLMs enjoy several potential benefits at inference time: they can generate text in arbitrary order, are naturally self-speculative (i.e. one can see the model's best guess for the entire sequence at every step), and can trade off compute and generation quality by choosing to unmask more or less tokens per step.

Here we discuss several unique characteristics of dLLM inference, which may be helpful to readers who are only familiar with traditional AR LLM inference.

**Fixed sequence length.** One drawback of dLLMs is that the context length must be fixed ahead of time, instead of being dynamically grown as with AR LLMs. Works such as BlockDiff address this issue by introducing a hybrid autoregressive/diffusion framework (Arriola et al., 2025); in this paper, we stay within the normal diffusion framework for simplicity.

**Instruct-tuned models.** dLLMs such as LLaDA-8B-Instruct, which have undergone supervised fine-tuning (SFT) on instruction-following traces, tend to place higher probabilities on `EOS` tokens. When combined with confidence-based unmasking, this leads to an unnaturally high proportion of `EOS` tokens in later positions and terse, stilted responses (Nie et al., 2026). Thus, to generate text with standard left-to-right prompting, we divide the response into smaller blocks, unmask tokens within the leftmost block, and continue to the next block once all tokens in the current block have been unmasked. This is known as *semi-autoregressive* sampling (Yang et al., 2025). During training, we use semi-AR sampling with block length 12.

**Bidirectional prompting.** In this paper, we work with traditional left-to-right prompting, which is the native format for the reasoning datasets we use. This leaves a big dLLM advantage on the table — namely their ability to generate text from arbitrary context. Future works could consider reasoning tasks that involve using context from both the left and right; for example, giving the model a problem and the numerical answer, and forcing it to deduce the intermediate steps.

**KV caching.** Since diffusion transformers typically use full self-attention instead of causal self-attention (Peebles & Xie, 2022), embeddings for the same token position can change over the sampling process. This non-causality prevents dLLMs

from using the same KV caching mechanism as AR LLMs. As a result, generating same-quality text with dLLMs is significantly slower than same-scale AR models, which is especially painful for online RL. However, there has been some recent interest in KV caching alternatives for dLLMs (Wu et al., 2025; Ma et al., 2025). Since dLLMs already have the ability to decode multiple tokens in parallel, we believe a successful implementation of KV caching is imperative to realizing dLLMs' potential as a faster, more flexible alternative to AR models.

## G. Sample Responses

See Table 4 for the system prompt used in GSM8K training as well as sample responses from a chosen test split problem. As evident in the responses, RL induces a rigid step-by-step structure, which we hypothesize allows the model to arrive at the same solution with many less steps ($m = 48$ vs $m = 192$).

Even though both responses arrive at the correct answer, the $m = 192$ solution is more coherent, while the $m = 48$ solution contains clear artifacts (i.e. repeated words, grammar mistakes). Note that the model is only trained on $m = 128$, so such artifacts are technically out-of-distribution and could ostensibly "derail" the model. Our results suggest that learned reasoning abilities are in fact robust to such perturbations.

*Remark* G.1. Overall, we observe that models trained with AGRPO can exhibit signs of entropy collapse, namely reduced token diversity and formulaic, "template"-y answers. This is consistent with findings in the broader RLVR literature. Future works could explore ways to mitigate entropy collapse by e.g. setting a higher clipping threshold in the clipped objective (Yu et al., 2025).

*Table 4.* Sample responses ($n = 384$) for different values of $m$ (# sampling steps) from a model trained with AGRPO. Problem is from the GSM8K-Platinum dataset.

---

**System:**

```
You will be given a math problem.
Think through the problem step by step, enclosing your reasoning process
in <think> tags.
Provide your final answer in <answer> tags, with only the number or
expression enclosed in \boxed{}, as follows:
<think>
...
</think>
<answer>
\boxed{...}
</answer>
```

**User:**

```
Tommy is fundraising for his charity by selling brownies for $3 a slice
and cheesecakes for $4 a slice. If Tommy sells 43 brownies and 23 slices
of cheesecake, how much money does Tommy raise?
```

**Assistant (m=48):**

```
<think>
First, we need to calculate the total from from selling selling brownies.
Each Eachie is $ $3 and and Tommy sells 43 brownies
\[\[
3 \times 43 =  = 129
\]
Next, we calculate the total from selling selling selling cheesecakes.
Each slice of cheesecake is $4 and and sells 23 slices is:
\[\[4 \times 23 = 92
\]
Finally, we add these two amounts to find the total money raised by
Tommy:
\[\[129 + 92 = 221
\]
</think>
<answer>
\boxed{221}
</answer>
```

**Assistant (m=192):**

```
<think>
First, we need calculate the money raised from selling the brownies.
Tommy sells 43 brownies at $3 each:
\[\[
43 \times 3 = 129
\]
Next, we calculate the money raised from selling the cheesecakes. Tommy
sells 23 slices of cheesecake at $4 each:
\[\[
23 \times 4 = 92
\]
Finally, we add these two amounts together to find the total amount of
money raised:
\[\[
129 + 92 = 221
\]
</think>
<answer>
\boxed{221}
</answer>
```

