# OpenReview forum: "Simple Policy Gradients for Reasoning with Diffusion Language Models"
_ICML.cc/2026/Conference — ICML 2026 regular_

### Official Review · Reviewer_Y86z · 2026-02-24

**Soundness:** 2
**Presentation:** 3
**Significance:** 3
**Originality:** 2
**Overall Recommendation:** 3
**Confidence:** 3

**Summary:**

This paper proposes AGRPO, a GRPO-based RL post-training algorithm for DLMs, treats the DLM generation process as an MDP, and leverages each decoding step’s output distribution as the policy. With a series of variance-reduction tricks adopted, AGRPO outperforms recent DLM post-training methods that leverage ELBO to approximate the sequence-level likelihood on multiple standard benchmarks.

**Compliance With Llm Reviewing Policy:**

Affirmed.

**Key Questions For Authors:**

Please see the weaknesses section.

**Limitations:**

Please see the weaknesses section.

**Strengths And Weaknesses:**

**Strengths**

- This work proposes an alternative view of RL for DLMs. Instead of the traditional approach of substituting the intractable likelihood with the ELBO approximation, AGRPO originates from the DLM generation process.
- The entropy importance sampling in Sec. 4.2.2 is a novel approach to reduce the RL training variance.
- The experimental results are promising. It achieves better accuracy compared with other ELBO-based methods.

**Weaknesses**

- Although the method is effective, I think the policy or joint distribution $\pi(o_t^i | q, o_t^{i-1})$ (starting from Eq. 4) is still intractable in DLM generation and could only be approximated by the multiplication of the factorized distribution of each token (implementation details are missing in the current manuscript). Although this is a very common practice in DLMs, the policy distribution for each DLM decoding step is still not theoretically tractable or exactly computable; if so, it would be inaccurate for the authors to claim that AGRPO achieves an ‘exact likelihood’.
- This work only uses a fixed number of sampling steps $m$ for RL training, and the sensitivity of this hyperparameter is less analyzed, which is also connected to the problem of how well the policy is approximated.
- The idea of treating each DLM decoding step as an action for policy optimization is similar to LLaDOU (NeurIPS 2025) [1], which should be an important baseline for this work, more comparisons with LLaDOU are necessary.

[1] Reinforcing the Diffusion Chain of Lateral Thought with Diffusion Language Models. https://arxiv.org/abs/2505.10446

---

> ### Author Rebuttal · Authors · 2026-03-28
>
> We thank the reviewer for the thoughtful feedback and for recognizing AGRPO’s **promising results and novel perspective,** including our entropy importance sampling technique. We address the reviewer’s concerns in more detail below.
>
> ### Weaknesses
>
> W1. *Tractability of the per-step policy.*
>
> The reviewer correctly identifies that the per-step joint distribution $\pi_\theta(o_t\mid q,o_{t-1})$ is factorized as a product of per-token marginals. Importantly, this assumption is **inherent to standard dLLM modeling and inference,** not an approximation that we introduce. More precisely, the “exactness” we claim is with respect to the stepwise policy induced by the model’s inference procedure, which assumes conditional independence across masked positions (Section 2.1). We **do not seek to address the tractability of the true joint distribution,** nor do we claim exactness with respect to it; rather, we define the MDP to align with the generation process itself (Section 3.3), which ensures that **the optimized policy and the rollout-generating policy are consistent.** This consistency gap is a key drawback of ELBO-based approaches to dLLM RL, since their training objective diverges from the actual inference procedure.
>
> We will clarify this scope explicitly in the revision and contrast it with ELBO-based methods, which assume a sequence-level MDP and seek to approximate the sequence likelihood.
>
> W2. *Sensitivity to fixed m.*
>
> Thank you for bringing up this concern; we agree that a deeper analysis would be beneficial. We note that in our setup, m controls both the denoising horizon length and the per-step action granularity (# of tokens unmasked=n/m). For the main training runs (Section 5.2), we fix a reasonable setting (n=384, m=128) that balances wall time and generation quality, following best practices in prior dLLM RL works.
>
> We conducted a sensitivity check on Countdown (n=192, k=24) and observed **consistent convergence across m=48 and m=96.** Please see the reward curve here (anon link): https://imgur.com/a/agrpo-training-m-ablation-UwKkmWc. The final results are given below:
> | Model | Countdown |
> | --- | --- |
> | LLaDA (m=48) | 17.6 |
> | LLaDA (m=96) | 20.3 |
> | AGRPO (m=48) | 55.9 |
> | AGRPO (m=96) | 63.3 |
>
> These results, together with Section 5.3.1, suggest that AGRPO’s reasoning gains are **robust across different denoising granularities** for both training and inference. We will include this analysis in the revision.
>
> W3. *Comparison to LLaDOU/DCoLT.*
>
> The reviewer is referring to the DCoLT paper, which we already discuss in **Appendix C.** We agree that this is a crucial baseline and will add direct comparisons in the Results section of the main text. The key distinctions are:
> * DCoLT requires training a **separate Plackett-Luce ranking module** to select unmasking positions and accumulates gradients **over all m timesteps,** increasing complexity and compute costs.
> * AGRPO **directly optimizes the native dLLM policy** without auxiliary training objectives, using unbiased MC estimation to reduce loss computation to O(k) << O(m), resulting in a simple setup that aligns with the underlying generative process.
>
> Empirically, on the same GSM8K setup (n=256, m=128, LLaDA + LoRA), AGRPO outperforms reported DCoLT results by **+3.3%**:
>
> | Model | GSM8K |
> | --- | --- |
> | DCoLT | 84.7 |
> | AGRPO | 88.0 |
>
> ### Summary
>
> We will (1) tighten the scope of the “exact” language to the stepwise policy induced by dLLM inference, (2) clarify that the factorized distribution defines the policy being optimized, (3) add an ablation studying the training-time tradeoff induced by m, and (4) elevate the DCoLT comparison to the main text with direct numbers.
>
> We hope these clarifications address the reviewer’s concerns.

---

> > ### Author Rebuttal · Reviewer_Y86z · 2026-04-01
> >
> > Thanks for your response.
> >
> > W1: not addressed. The stepwise policy is indeed not an exact likelihood, so directly leveraging it in the policy optimization framework is an approximation, more formal theoretical analyses on the approximation error might be required.
> >
> > W2: addressed.
> >
> > W3: not addressed. 1) The evaluation is limited, only containing GSM8K, and in the LLaDOU paper, the GSM8K accuracy is 88.1%, which is much higher. 2) I think the unmasking policy module in LLaDOU is optional, so the method is still quite similar.

---

> > > ### Author Response · Authors · 2026-04-06
> > >
> > > We thank the reviewer for their attention and for marking W2 as resolved. To address the remaining concerns:
> > >
> > > W1. For more detail, please see our response to Reviewer NyeX’s W2 here: https://openreview.net/forum?id=89aKcIeUyP&noteId=wI0gx7nrLg. To summarize, while we agree that the stepwise policy is not “exact” relative to the CTMC/any-order AR formulation, we **do not** rely on this “exact” formulation for generating rollouts. Since the training and rollout policies are both defined by **the same m-step denoising process** (we use $m=n/3$), there is no inconsistency from the PG perspective.
> > >
> > > Another way to view it is that the PG objective only depends on action likelihoods $\pi(a\mid s)$ along the trajectory. By taking the action space $a$ to be 3 tokens instead of 1 token, we still get a valid policy $\pi$ (with a well-defined distribution over actions) which can be optimized without relying on single-token/CTMC likelihoods. PG **does not explicitly require** that the action space is 1 token, i.e. $m=n$, merely that the policy is **consistent between training and inference.** The granularity of the action space ($n/m$) can be considered a hyperparameter controlling the quality/wall-time tradeoff, analogous to the chunk size for AR models with multi-token prediction.
> > >
> > > We apologize if it was unclear that the inference policy uses $m=n/3$, and we will make explicit that we design AGRPO for **aligned training and inference policies** (a key gap in existing ELBO-based approaches) without depending on the “exact” any-order inference procedure.
> > >
> > > W3. Please note that 88.1% GSM8K accuracy reported for LLaDOU (Table 2 in their paper) is trained on a larger GSM8K + MATH dataset, and is **evaluated with the full $n=m=256$ decoding process,** which makes it difficult to compare with our reported $m=n/2$ accuracies. The LLaDOU paper includes a direct comparison to d1-LLaDA in Appendix C.2.3, using the same GSM8K training split and $m=n/2$ setup, which is where we report the 84.7% accuracy from. In this controlled setting, AGRPO outperforms LLaDOU by +3.3%.
> > >
> > > Regarding LLaDOU’s unmasking policy module, Section 4.2.3 in their paper shows the GSM8K accuracy is quite sensitive to changes in the unmasking policy, which suggests this is a purposeful design choice rather than optional.
> > >
> > > Regarding similarity, we agree that both AGRPO and DCoLT view denoising steps as intermediate reasoning steps. Our main contributions are the novel MC estimation framework, which drastically reduces the compute cost per loss from O(m) to O(k), along with principled variance-reduction techniques, which are **empirically effective** while remaining **unbiased.** DCoLT does not solve this key compute bottleneck, which significantly limits its scalability in the long-horizon/high-m regime.
> > >
> > > ---
> > >
> > > We hope these responses clarify our contributions, especially the **training/inference consistency** defined via the underlying MDP, which comparable ELBO-based methods lack, and our **tractable, unbiased framing of PG via MC estimation.** Thank you again for your time and valuable feedback!

---

### Official Review · Reviewer_NyeX · 2026-03-08

**Soundness:** 2
**Presentation:** 2
**Significance:** 1
**Originality:** 1
**Overall Recommendation:** 3
**Confidence:** 4

**Summary:**

This paper studies RL post-training for dLLMs, a promising alternative to AR LLMs. It proposes a variant of GRPO that is based on the Markovian nature of dLLM sampling process, and demonstrates its effectiveness on a variety of benchmarks.

**Compliance With Llm Reviewing Policy:**

Affirmed.

**Final Justification:**

My initial evaluation that the paper is like a graduate-level course project may be too harsh and I sincerely apologize for this.
During the rebuttal, the authors have made efforts to resolve many of my concerns and questions. In light of this, I decided to increase the rating from 2 to 3.
However, I still don't feel fully convinced by the claimed contribution of this work. I see their estimator is unbiased in some sense and doesn't involve biased estimate by ELBO, but from my understanding this is not outstandingly novel improvement over the existing methods for fine-tuning dLLM.
Moreover, since modification of the paper during the rebuttal is not allowed, many of the promised changes, new results, and the authors' possible improvement in writing during the revision, would definitely need further thorough review for a more prudent decision. Therefore, I don't incline toward acceptance of this submission.

**Key Questions For Authors:**

1. For drawing $k$ samples of $T\sim\\{1,...,m\\}$, I don't quite understand the intuition behind the use of low-discrepancy sampling here. Why don't we just draw a uniformly random subset of size $k$ from $\\{1,...,m\\}$ (i.e., without replacement)? Repetitions of the same $T$ value don't seem to be nice here. Finally, could the authors explain the barrier of using all $m$ values? Is it due to memory constraints?
2. Let's go back to the most fundamental question: why do we want to use $\mathcal{J}(\theta)$ as the objective function? The authors spent a lot of effort in the paper to justify that their choice of the estimator is unbiased of $\mathcal{J}(\theta)$, but I don't see a clear justification for why we choose this in the first place. In papers like GSPO (https://arxiv.org/abs/2507.18071) and GMPO (https://arxiv.org/abs/2507.20673), people found that the original PPO objective is not a good surrogate in the sequence level.
3. Could the authors also include the ablation study on: (1) the block length of generation and the block length of splitting the loss (note that they need not be the same); (2) the frequency of resampling the buffer under $\pi _ \mathrm{old}$; and (3) the number of batch size of rollouts per prompt?
4. Does $\beta=0$ (no KL regularization) suffer from reward hacking?
5. Could the authors explain in more detail what's the custom inference engine they used for LLaDA?

Minor comments:

6. Line 193: $o _ 0$ is typically an empty set?
7. Please check the formatting of the references especially the Deepseek papers. The author list is too long and should be truncated with "et al." after the first few authors. Also, some URLs are not properly broken into multiple lines.

**Limitations:**

The authors have not discussed the limitations. See the weakness part above.

**Strengths And Weaknesses:**

## Strengths

- RL for dLLMs is an important and timely topic, and the paper is well-motivated.
- The proposed method is well-positioned in the context of policy gradient and GRPO methods, and the paper provides a detailed explanation of it.
- The experimental results show improvements compared to baseline methods.

## Weaknesses

- The paper spent much space on the background knowledge and detailed derivation of the method. Much of the preliminary knowledge is already well-known to the community and is not directly used in the proposed method, while much of the experimental setup and literature review can be put to the appendix. I believe the paper could be more concise and informative if it focuses more on the core method.
- I don't see too much novelty in the proposed method. The final algorithm is simply **block-wise GRPO**, which replaces the per-token probability ratio with a per-block ratio. In particular, if the block length is chosen as one, then this is exactly GRPO. Therefore, although the authors claimed in remark 4.1 that their objective is from a fundamentally different assumption compared to GRPO, I don't see much difference between the two methods.
- Although the authors replaced the ELBO approximation used in d1 and related works, if I understand correctly, the final algorithm still relies on the approximation of joint likelihoods as the product of per-token likelihoods: $\pi _ \theta(o _ t|q,o _ {t-1})\approx\prod _ {i~\text{in the block}}\pi _ \theta(o _ {t,i}|q,o _ {t-1})$ (also $\pi _ \theta(o _ t|q,o _ {t-1})$ is not quite a good notation for dLLMs). This is not the true conditional likelihood of the newly unmasked block given the previous blocks under the dLLM, so I suggest a change in the PPO surrogate objective (4), which makes the unbiasedness claim more accurate.
- For the experiments, I'm curious why the authors don't compare on generation length 128 as done in previous works. Also, I suggest the authors to include more recent baselines such as DiffuCoder (https://arxiv.org/abs/2506.20639), d2 (https://arxiv.org/abs/2509.21474), DMPO (https://arxiv.org/abs/2510.08233) and GDPO (https://arxiv.org/abs/2510.08554), at least for discussion in the related work section.
- The performance of variance reduction is not properly justified. The gradient norms are on average smaller than baseline, but have multiple spikes, which on the contrary may indicate instability. A more fair comparison would be to directly report the variance of the gradient estimator on a fixed batch of samples.

Overall, due to limited novelty and lack of significant contributions, I think the paper is more like a graduate-level course project rather than a research paper worthy of publication in a top-tier machine learning conference.

---

> ### Author Rebuttal · Authors · 2026-03-27
>
> We thank the reviewer for the detailed feedback and for recognizing our work as **important and well-motivated.** We address several characterizations in detail below.
>
> ### Weaknesses
> 1. *AGRPO is simply block-wise GRPO.*
>
> While GRPO and AGRPO share superficial similarities (both deriving from the same PG objective), they arise from fundamentally different assumptions. GRPO starts from a **sequence-level** objective and relies on likelihood decomposition via AR chain rule, whereas AGRPO starts from a **multi-step denoising MDP**, where actions are the n/m tokens unmasked at a single denoising step. AGRPO’s IS ratios $\frac{\pi_\theta(o_t\mid q,o_{t-1})}{\pi_{old}(o_t\mid q,o_{t-1})}$ represent the marginals from the **t-th unmasking step** (conditioned on the partially masked state), whereas GRPO’s token IS ratios are derived from AR factorization of the sequence-level likelihood $\pi(o\mid q)$ (see Q2).
>
> Surface-level similarities aside, our core contribution is making this objective **tractable for dLLMs** without ELBO surrogates. Fully computing the sequence-level objective requires either O(m) forwards or AR factorization; our method obtains an unbiased estimate with only O(k) forwards via timestep subsampling, which is only available using stepwise likelihoods along the denoising trajectory.
>
> 2. *Joint approximated via product of per-token marginals.*
>
> The reviewer is correct about this joint factorization. We want to be precise: this is not an approximation we introduce, but rather the **standard dLLM inference procedure itself.** dLLMs are trained to model independent marginals over masked positions, and this product factorization carries over to the sampling process. When we define the policy/MDP to align with the sampling process (Section 3.3), we naturally inherit this factorization.
>
> Regarding the RL objective, we define the policy at the granularity of denoising steps (n/m tokens), **not individual tokens.** Accordingly, our unbiasedness claim is w.r.t. this induced policy (the same policy used at inference), not the intractable joint distribution. We will clarify this point in the revision.
>
> 3. *Missing baselines (seq len 128, DiffuCoder, d2, DMPO, GDPO)*
>
> We are happy to include n=128 results in Table 1 in the revision. Preliminary results show consistent gains in this setting:
>
> | Model | GSM8K | Countdown |
> | --- | --- | --- |
> | LLaDA | 70.6 | 16.4 |
> | AGRPO | 78.6 | 67.2 |
>
> Regarding baselines, DiffuCoder is pre- and post-trained for the coding setting instead of mathematical reasoning, making direct comparison difficult. d2, DMPO, and GDPO are recent works which were not included in our submission due to timing; we will include them as related works with in-depth discussion (Appendix C) in the revision.
>
> 4. *Variance reduction ablation, gradient norm spikes*
>
> Thank you for this suggestion! We agree that directly comparing gradient variance on a fixed batch of rollouts would be more rigorous and will include this in the revision. The isolated gradient norm spikes in Figure 4 do not materially affect optimization; training converges stably and consistently across seeds for both runs.
>
> ### Questions
>
> Q1. Sampling uniformly w/o replacement is indeed a valid, unbiased estimator. We use low-discrepancy sampling due to its stronger coverage guarantees: it partitions all timesteps into k equal-size bins and samples one from each. This helps the model learn from the **full range of intermediate states**, reducing the chance of sampling clustered timesteps. Regarding using all timesteps: with ~8B models and m=128 steps, running m (instead of k << m) forwards per gradient step would explode the loss/backprop cost, making training prohibitively slow (Figure 3(b)).
>
> Q2. GSPO/GMPO assume a **one-step MDP** with a sequence-level policy, where they argue sequence-level IS ratios are more natural and stable. This idea of **aligning the objective with the underlying MDP** actually supports our choice: under our **multi-step MDP,** the natural unit of action is a single denoising step, so the standard PG objective (4) naturally yields step-level IS ratios. We are thus not subject to the same critique of token-level ratios (see also [1]), and our strong empirical gains over ELBO-based approaches further validate this choice.
>
> Q3-7. We are constrained by space but are happy to discuss this further if the reviewer feels these would considerably affect their assessment.
>
> In summary, AGRPO’s key contributions — a principled, unbiased PG framework aligned with dLLM inference, free of ELBO surrogates, with concrete variance reduction techniques and empirical improvements over all prior dLLM RL methods — go well beyond what the “course project” characterization implies. We will update the results with n=128 and address the presentation gaps in the revision.
>
> [1] Zheng et al., 2025. Stabilizing Reinforcement Learning with LLMs: Formulation and Practices. https://arxiv.org/abs/2512.01374

---

> > ### Author Rebuttal · Reviewer_NyeX · 2026-04-03
> >
> > I thank the authors for the reply. Due to the word limit, there are several questions and concerns that the authors have not yet fully responded to. I will be willing to learn from their follow-up reply to see if I can potentially revise my rating. However, generally speaking, I don't feel the rebuttal has changed my opinion much. Even after the promised revisions as indicated by the authors, the paper is still not strong enough for publication in the top-tier machine learning conferences from my understanding.
> >
> > Regarding W1 and W2 (methodological concerns), I'm not fully convinced by the authors' explanation.
> >
> > W1: If you treat each step as denoising a block of $\ell$ tokens at once, then the derivation inherits from GRPO. The only difference is that $\ell$ is not one in this case. On the other hand, AR generation for typical AR LLMs is just unmasking one position at each denoising step.
> >
> > W2: I disagree with the claim that this product approximation is the standard dLLM inference procedure. True, dLLMs learn the per-token conditional probabilities, but the **exact** inference (from the perspective of CTMC, or equivalently random-order AR generation, see the RADD paper https://arxiv.org/abs/2406.03736 for instance) is to sample a uniformly random order and autoregressively sample each position conditional on the previously sampled ones, one by one. The typical inference procedure (which you use for the rollouts, as indicated in line 312-314) does not unmask the whole block at once, but with multiple unmasking steps. Therefore, when you compute the IS ratio for the whole block, you are actually using the product of the per-token IS ratios rather than the true sampling probability. (The reviewer Y86z's W1 is also about this issue)

---

> > > ### Author Response · Authors · 2026-04-05
> > >
> > > We thank the reviewer for their continued engagement and careful follow-up. We address the remaining concerns below:
> > >
> > > ### Questions
> > >
> > > Q3. For (1), please see the ablation on $m$ in our rebuttal to Reviewer Y86z (W2): https://openreview.net/forum?id=89aKcIeUyP&noteId=F1vDrEKPcO. The key finding is that performance is **robust across a range of $m$ values.** We note that decoupling the value of $m$ used for rollout vs. loss computation introduces a train/inference mismatch (see W2 below), so we recommend keeping them consistent.
> > >
> > > For (2), we ran an ablation on Countdown using $\mu=1$ (fully on-policy) and $\mu=4$. The reward curves are here (anon link): https://imgur.com/a/agrpo-mu-ablation-L1ZzHm2 (The larger reward variance for $\mu=4$ is expected since rewards are calculated for new prompts, and there are fewer new prompts per rolling window.)
> > >
> > > Final results:
> > > | Model | Countdown (n=192, m=96) |
> > > | --- | --- |
> > > | LLaDA | 20.3 |
> > > | AGRPO (mu=1) | 63.3 |
> > > | AGRPO (mu=4) | 61.3 |
> > >
> > > The minimal gap between $\mu=1$ and $\mu=4$ suggests that AGRPO training is **robust, even with moderate levels of buffer replay.**
> > >
> > > For (3), the rollouts per prompt is fixed at $G=8$ with 16 prompts per update (Appendix B.1), which is standard for LLM post-training works (GRPO, DAPO, d1, etc.).
> > >
> > > Q4. We don’t observe notable reward hacking with $\beta=0$, consistent with previous RL works (wd1, DAPO) which find KL regularization unnecessary for math/coding tasks.
> > >
> > > Q5. The custom inference engine extends the standard LLaDA sampler by caching partially masked states, log-probs/entropies, and top-p/top-k masks (Appendix E), which we reuse when computing the loss. We are happy to release the code publicly after the review period.
> > >
> > > Q6. Yes, $o_0$ is the all-masked state, which is concatenated with the prompt $q$.
> > >
> > > Q7. Thank you! We have fixed this formatting issue.
> > >
> > > ### Weaknesses
> > >
> > > W1. We agree that AGRPO, GRPO, and many other LLM RL works all share the same underlying PG foundation. This is expected, since the PG approach has proven highly effective. Our goal is therefore not to propose an entirely new PG formulation, but to make this formulation **tractable for dLLMs** — without any ELBO-based surrogates — by reconsidering the underlying MDP. The key contribution is the **unbiased MC estimator** that relies on **intermediate denoising steps.** This framing naturally aligns with dLLMs, and also enables our variance-reduction techniques.
> > >
> > > W2. Indeed, the factorized probabilities from the m-step sampling process can be viewed as an approximation for the “true” probabilities under the CTMC/random-order process. Our claim is that AGRPO is unbiased with respect to the **actual m-step policy used to generate rollouts.**
> > >
> > > A key clarification is that we **do not use** $m=n$ when generating rollouts. As stated in Section 5.2, we use $n=384$ and $m=128$ (with semi-AR generation, block length=12, so each block still takes multiple steps to unmask) **for both rollouts and the training objective.** Generating rollouts with $n/m>1$ is common practice for dLLMs to balance wall time and coherence (d1 and SPG both use $n/m=2$). We will make this point explicit in the revision to avoid any confusion.
> > >
> > > We therefore **do not** claim equivalence to the “exact” random-order procedure that the reviewer mentions. Rather, we define the MDP so that the likelihoods we use in training are consistent with the **stepwise** likelihoods under the m-step rollout policy. By contrast, ELBO-based methods optimize a different sequence-level surrogate, introducing a mismatch with the actual rollout-generating distribution. AGRPO eliminates this train/inference gap by construction.
> > >
> > > More broadly, our MDP framework is defined for general $m$ (L120), including the CTMC/any-order procedure as a special case $m=n$. This means that our PG algorithm **does not** depend on the “exact” CTMC/any-order likelihoods. The crucial point is that **as long as $m$ is consistent between training and inference, the action spaces are aligned,** and one is not an approximation of the other.
> > >
> > > ### Summary
> > >
> > > We hope the above points — particularly that rollouts use $m=n/3$, and that “exactness” should be judged relative to the rollout-generating process — clarify the main remaining disagreements. To briefly restate our contributions for completeness: (1) a tractable, unbiased MC estimator for PG that requires no ELBO surrogates, (2) train/inference alignment via consistently defined action spaces ($m$) between rollouts and loss, (3) principled variance-reduction techniques for dLLM reasoning (low-discrepancy sampling and entropy IS), and (4) consistent empirical improvements over comparable methods across standard benchmarks. We believe that, taken together, these represent a meaningful contribution for dLLM RL.
> > >
> > > Once again, we thank the reviewer for their thorough feedback, and we hope the reviewer and AC consider these contributions in their final assessment.

---

### Official Review · Reviewer_hmKD · 2026-03-13

**Soundness:** 4
**Presentation:** 4
**Significance:** 4
**Originality:** 4
**Overall Recommendation:** 6
**Confidence:** 4

**Summary:**

This papers proposes AGRPO for training diffusion LLM by considering the right setup as multi step MDP where the state is partially masked sequence, actions are token to be unmasked and reward at terminal step. This consideration in the MDP moves the focus from sequence likelhood attempted in previous work (that is intractable ), to the stepwise action log-likelihood of unmasked token given partially masked sequence at time t (that is computable). This allows under this natural multi-step MDP for dLLM to write a GRPO like algorithm using these log likelhoods and importance sampling on these likelihoods.  The inference happens on pi_{\theta_{old}} using m steps of unmasking and we obtain G rollouts.  For each, given m steps in diffusion process, the estimator samples k time steps in {1,..m} in the unmasking process  to estimate GRPO's advantage. Another techniques for sampling diffusion timesteps are proposed in Section 4.2 to reduce variance. Efficiency of the method is demonstrated on reasoning tasks with convincing experimental result.

**Compliance With Llm Reviewing Policy:**

Affirmed.

**Final Justification:**

I keep my positive assessment of the paper and recommend acceptance

**Key Questions For Authors:**

My main question is on Section 4.2 using sampling that depend on the policy breaks a bit the policy gradient and maybe need a particular care and analysis ? is the main goal of introducing the variance reduction to use smaller k ?

**Limitations:**

It would be nice to add in Table 1 variances.

**Strengths And Weaknesses:**

* Soundness/ original: This is a simple yet clever and elegant paper, I enjoyed reading it, and finding the right probability space on which to define the mdp that is natural is interesting and original to the best of my knowledge.

* presentation: the presentation of the paper is also good and the paper is easy to follow.

* Significance: The paper is a nice contribution for RL training of diffusion LLMs.

---

> ### Author Rebuttal · Authors · 2026-03-29
>
> We thank the reviewer greatly for the thorough reading and generous assessment. We are glad that the paper’s core idea — “finding the right probability space on which to define the MDP” — came across as **elegant and well-motivated,** and that the reasoning experiments yielded **convincing results.** We address the three main questions below.
>
> Q1. *Does policy-dependent sampling break the policy gradient?*
>
> Thank you for raising this concern. The key property is that both variance reduction techniques (low-discrepancy sampling and entropy IS) **preserve unbiasedness of the gradient estimator** with respect to the induced stepwise policy $\pi_\theta$ (see Appendix A).
>
> Regarding entropy importance sampling: the entropy scores $e_t$ are computed under $\pi_{old}$ (the rollout-generating policy), not the current $\pi_\theta$ being optimized. When computing the entropy IS objective, the $e_t$ terms are wrapped with a stop-gradient operation (`.detach()` in PyTorch). This makes them **constant with respect to $\theta$,** resulting in a standard importance-weighted MC estimator for $\mathcal J(\theta)$.
>
> Intuitively, we are simply re-weighting which timesteps to MC sample via a fixed importance measure, not altering the actual denoising MDP or PG update itself.
>
> Q2. *Is the goal of variance reduction to use smaller k?*
>
> Yes, one of the primary goals of the variance reduction techniques introduced in Section 4.2 is to maintain gradient quality with fewer MC samples k. With low-discrepancy sampling, we ensure that our samples cover the full timestep range {1,...,m}, reducing the risk of redundancy from closely sampled timesteps. With entropy IS, we ensure that our samples are concentrated on the most high-entropy/informative timesteps. Together, these enable efficient, scalable training with k << m while maintaining stability and unbiasedness. Our ablations (Section 5.5) further validate these variance reduction techniques.
>
> Q3. *Can you show variances in Table 1?*
>
> Thank you for this suggestion. Our evaluation uses a deterministic inference procedure (confidence-based unmasking with temperature 0), following conventions in prior dLLM works. While large-scale RL runs in prior works are typically reported with a fixed seed due to the costly nature of full-attention dLLM training, we agree that training-time variance (from rollout sampling, MC estimation, and LoRA initialization) is relevant. We are happy to add a smaller ablation on Countdown performance across different seeds in the revision.
>
> ### Summary
> We hope the above clarifications fully address the reviewer’s concerns. As the reviewer notes, we believe AGRPO’s principled formulation, tractable training recipe, and strong empirical gains represent a meaningful step forward for dLLM post-training.

---

> > ### Author Rebuttal · Reviewer_hmKD · 2026-04-01
> >
> > Thanks for the clarifications, I still think the paper is very interesting and agree that the method is " meaningful step forward for dLLM post-training."

---

### Official Review · Reviewer_SXRR · 2026-03-13

**Soundness:** 3
**Presentation:** 2
**Significance:** 2
**Originality:** 2
**Overall Recommendation:** 3
**Confidence:** 4

**Summary:**

This paper proposes AGRPO, a reinforcement learning method for diffusion language models based on a multi-step MDP reformulation. The central idea is to replace sequence-level likelihood surrogates with stepwise token/action likelihoods defined along the denoising trajectory, and to estimate the sum over denoising steps via Monte Carlo timestep sampling. Empirically, the method is evaluated mainly on LLaDA-style diffusion language models and compared against several prior dLLM RL baselines.

**Compliance With Llm Reviewing Policy:**

Affirmed.

**Key Questions For Authors:**

1. Please clarify the exact scope of the “exact” claim. Is the claim intended to apply only to stepwise likelihoods conditioned on the selected unmasking positions / partially masked state, or to the full inference procedure including position selection?
2. How should one interpret the theory when confidence-based unmasking is used, since the selected positions then depend on the model itself?
3. Do the authors have evidence that the timestep-sampling strategy remains sample-efficient in very long denoising trajectories, where critical reasoning steps may be sparse?
4. Why were newer trace-aware dLLM RL baselines such as TraceRL and DiRL/DiPO not included?
5. Can the authors comment on whether AGRPO is expected to transfer cleanly to block diffusion / hybrid diffusion architectures such as SDAR[1] and fast-dllm-v2[2]?

[1] SDAR: A Synergistic Diffusion-AutoRegression Paradigm for Scalable Sequence Generation
[2] Fast-dLLM v2: Efficient Block-Diffusion LLM

**Limitations:**

The paper discusses computational tradeoffs, but I think the limitations should be made more explicit in two places. First, the theoretical claim appears to depend on treating the unmasking rule as fixed/exogenous, which is less clean when confidence-based unmasking is used. Second, the empirical study does not yet show whether the timestep-sampling estimator remains effective in very long-trace reasoning settings or in newer blockwise diffusion architectures.

**Strengths And Weaknesses:**

**Strengths**

- The paper addresses an important problem: how to perform RL for diffusion language models without relying on sequence-level likelihood surrogates that are often poorly aligned with the actual decoding process.
- The stepwise reformulation is conceptually meaningful, and the paper makes a reasonable effort to connect the method to policy-gradient style training.
- The empirical results are also generally competitive against the included baselines, and the paper is fairly clear about the computational motivation for sampling a subset of denoising steps instead of explicitly summing over the full trajectory.

**Weaknesses**
- The paper’s “exact” wording is too strong. What seems justified is an unbiased MC estimator for the token-step PG objective under the paper’s modeling assumptions, not exactness for the full inference procedure. In particular, the footnote treats unmasking positions as fixed/exogenous and “not explicitly optimized,” while also grouping random and confidence-based unmasking together. This may be acceptable for random schedules, but it is much less clear for confidence-based unmasking, where position selection depends on the model itself. In that case, the claimed exactness appears to hold only conditionally on the selected positions / partially masked state, rather than obviously for the full decoding mechanism.
- The efficiency / credit-assignment tradeoff for long traces. AGRPO replaces the full sum over denoising steps with MC sampling over (k) timesteps. This is principled, but in long-CoT or long-trace settings it is unclear whether a small (k) can reliably cover the rare but critical steps that determine the final outcome. Increasing (k) may help, but also increases the number of forwards and training cost. The current experiments and ablations are informative, but they do not yet establish that the method remains efficient and stable when the denoising horizon becomes very long.
- The baseline set feels incomplete relative to the current dLLM RL literature. Since the main claim is about principled and tractable trajectory-level RL for diffusion models, I would expect comparisons to newer trace-aware methods such as TraceRL and DiRL/DiPO, as well as other trajectory/schedule-aware approaches such as MDPO, in addition to ELBO-style baselines.
Finally, the experiments are concentrated on full-attention LLaDA-style diffusion models. Given the field’s shift toward blockwise / hybrid diffusion architectures, it would strengthen the paper to test whether the proposed objective and estimator remain useful in that regime as well.

---

> ### Author Rebuttal · Authors · 2026-03-27
>
> We thank the reviewer for the thorough and helpful feedback. We are glad that the reviewer recognizes AGRPO’s clear motivation and empirical strength, and that **“the stepwise reformulation is conceptually meaningful.”** We address some of the main concerns below.
>
> ### Weaknesses
> 1. *“Exact” wording is too strong.*
>
> Thank you for raising this concern. Our usage of “exact” is scoped to stepwise action likelihoods, i.e., the exact likelihood of a **single unmasking step** conditioned on unmasking positions and partially masked state, not exactness with respect to the full generation process. This is mainly to contrast with ELBO-based approaches which treat the **entire sequence** as the action, forgoing a tractable, exact likelihood (cf. Figure 1). We will clarify throughout the paper (L146, L803) that the “exact” refers to “exact stepwise likelihood conditioned on partially masked state and unmasking schedule.”
>
> 2. *Confidence-based unmasking changes MDP assumptions.*
>
> Indeed, confidence-based unmasking introduces a dependency on the policy for unmasking positions. All training is conducted with random unmasking, fully aligning with our proposed MDP. Confidence-based unmasking is used only at evaluation to ensure a **fair comparison with prior works** that report results under this setup (L323).
>
> While we acknowledge this is technically a train-test mismatch, we view confidence-based schedules as a form of annealing that produces more coherent generations than random schedules, analogous to training AR models via sampling and evaluating with beam search.
>
> To give additional context, we report random unmasking results from GSM8K and Countdown below:
> | Model | GSM8K | Countdown |
> | --- | --- | --- |
> | AGRPO (confidence) | 88.0 | 79.7 |
> | AGRPO (random) | 83.8 | 80.0 |
> | LLaDA (confidence) | 79.9 | 20.3 |
> | LLaDA (random) | 66.6 | 11.7 |
>
> These accuracy gaps are consistent with the well-studied empirical benefits of confidence-based schedules. Notably, AGRPO shows strong relative gains over the base LLaDA model under both schedules.
>
> 3. *Sample efficiency and stability for long denoising horizons.*
>
> We agree that this is a limitation. Current full-attention dLLMs (like LLaDA) are generally impractical for long-context settings, so we were unable to run extremely long-horizon experiments, a constraint shared by all prior dLLM RL works. However, we expect AGRPO to extend smoothly to longer horizon tasks, since MC estimation breaks the fundamental one-to-one dependence between # forwards and denoising steps. In addition, our proposed **entropy importance sampling** specifically targets high-entropy states, reducing the risk of missing rare but critical reasoning steps.
>
> Our ablations (Section 5.4) show consistent improvements as k increases from 2 to 32, suggesting coverage improves gracefully. (In high m regimes, increasing k can increase the # of forwards per step, but can also decrease the total # of steps, thus reducing the total compute cost.) We will explicitly flag very long denoising trajectories as an open question in the revision.
>
> 4. *Incomplete baseline set.*
>
> Thank you for bringing these dLLM RL works to our attention. We agree that trajectory-based RL baselines would strengthen the paper, and will incorporate direct comparisons with TraceRL and MDPO. To our knowledge, AGRPO remains the only trajectory-based RL framework that uses **unbiased estimation** to make PG updates tractable for **pure diffusion language models.** To elaborate briefly:
> * TraceRL’s “shrinkage parameter” compresses a trajectory of m steps into m/s steps to make PGRL feasible. However, this shrinkage changes the granularity of the MDP, introducing a train/inference mismatch, which AGRPO avoids via unbiased MC estimation.
> * MDPO introduces process supervision via intermediate answers. It requires **m forward passes** for a m-step trajectory, significantly limiting MDPO’s scalability for longer time horizons.
> * DiRL relies on a **block diffusion structure** with simplified likelihood computation. Their main focus is architectural efficiency/training latency. (DiRL was released publicly in December 2025 and is concurrent by ICML policy.)
>
> ### Questions
> 1-2. See responses to Weaknesses 1-2.
>
> 3. See Weakness 3. We will add an explicit discussion of LLaDA’s context limitations in the paper.
>
> 4. See Weakness 4.
>
> 5. Yes, we expect AGRPO to transfer cleanly to block-diffusion models such as SDAR and Fast-dLLM-v2, since the decoding process still falls under our denoising MDP framework. The key techniques we introduce (timestep MC estimation, low-discrepancy, entropy IS) remain applicable without modification. We have not yet empirically validated this and will clarify it as a hypothesis rather than a demonstrated result.
>
> In summary, we will tighten “exact” claims, add random unmasking results, and include comparisons with TraceRL and MDPO. We hope these address the reviewer’s main concerns.

---

> > ### Author Rebuttal · Reviewer_SXRR · 2026-04-05
> >
> > Thank you for your rebuttal. However, concerns remain about the limited dataset and the need for experiments on more advanced models. Therefore, further revision is required before the next review round.

---

> > > ### Author Response · Authors · 2026-04-06
> > >
> > > We thank the reviewer for the timely response and concrete feedback.
> > >
> > > While we agree that further experiments with more advanced models would provide additional insights, doing so would require a significantly expanded compute budget. As other reviews have noted, our current set of experiments and ablations are already strong, and **clearly showcase AGRPO’s improvements over baselines.** The four datasets we use (GSM8K, MATH, Countdown, Sudoku) are standard reasoning benchmarks — the same ones used in prior dLLM RL works such as d1, SPG, etc. — and we show **consistent empirical gains** across all four tasks.
> > >
> > > ---
> > >
> > > To summarize, the remaining concerns appear to be about empirical scope (additional datasets/newer architectures), rather than the core methodological concerns raised in the original review. We have addressed those earlier concerns by clarifying (1) the scope of our claims with respect to the stepwise training/inference policy and (2) the role of the unmasking schedule.
> > >
> > > We hope this holistic view will be taken into account in the reviewer and AC’s final assessment. Thank you again for your valuable time and attention!

---

### Decision · Program_Chairs · 2026-04-30

**Decision:**

Accept (regular)

**Comment:**

This paper proposes AGRPO, a reinforcement learning framework for diffusion language models (dLLMs) that reformulates the denoising process as a multi-step Markov Decision Process. By leveraging Monte Carlo timestep sampling, the method provides an unbiased policy gradient estimator that significantly reduces computational complexity from $O(m)$ to $O(k)$, addressing a major bottleneck in dLLM post-training. While the initial scores were diverse, with several reviewers raising concerns regarding the "exactness" of the likelihood claims and the empirical scope, the Meta-Reviewer has carefully read the author response and the subsequent discussions. The rebuttal successfully clarified that the "exactness" refers to the principled alignment between the training objective and the actual $m$-step inference policy used during rollouts, effectively resolving the primary technical concern shared by the skeptical reviewers. A strong champion reviewer highlighted the elegance of the MDP formulation and the originality of the probability space used for the derivation. Although some reviewers maintained that the method represents an incremental extension of existing policy gradient methods, the Meta-Reviewer finds that the transition from sequence-level ELBO approximations to a tractable, unbiased trajectory-level objective is a significant contribution to the non-autoregressive modeling community. Given the robust performance gains on reasoning tasks and the potential for this framework to enable more efficient dLLM scaling, the paper is recommended for acceptance.